# Randomised trials conducted using cohorts: a scoping review

Beverley Jane Nickolls ![ORCID],[1] Clare Relton,[1] Lars Hemkens ![ORCID],[2,3,4,5] Merrick Zwarenstein,[6,7] Sandra Eldridge,[1] Stephen J McCall ![ORCID],[8,9] Xavier Luke Griffin ![ORCID],[10,11] Ratna Sohanpal,[12] Helena M Verkooijen,[13,14] Jonathon L Maguire ![ORCID],[15] Kimberly A McCord[16]

For numbered affiliations see end of article.

**Correspondence to**
Beverley Jane Nickolls;
b.nickolls@qmul.ac.uk

## ABSTRACT

**Introduction** Cohort studies generate and collect longitudinal data for a variety of research purposes. Randomised controlled trials (RCTs) increasingly use cohort studies as data infrastructures to help identify and recruit trial participants and assess outcomes.

**Objective** To examine the extent, range and nature of research using cohorts for RCTs and describe the varied definitions and conceptual boundaries for RCTs using cohorts.

**Design** Scoping review.

**Data sources** Searches were undertaken in January 2021 in MEDLINE (Ovid) and EBM Reviews—Cochrane Methodology Registry (Final issue, third Quarter 2012).

**Eligibility criteria** Reports published between January 2007 and December 2021 of (a) cohorts used or planned to be used, to conduct RCTs, or (b) RCTs which use cohorts to recruit participants and/or collect trial outcomes, or (c) methodological studies discussing the use of cohorts for RCTs.

**Data extraction and synthesis** Data were extracted on the condition being studied, age group, setting, country/continent, intervention(s) and comparators planned or received, unit of randomisation, timing of randomisation, approach to informed consent, study design and terminology.

**Results** A total of 175 full-text articles were assessed for eligibility. We identified 61 protocols, 9 descriptions of stand-alone cohorts intended to be used for future RCTs, 39 RCTs using cohorts and 34 methodological papers. The use and scope of this approach is growing. The thematics of study are far-ranging, including population health, oncology, mental and behavioural disorders, and musculoskeletal conditions.

Authors reported that this approach can lead to more efficient recruitment, more representative samples, and lessen disappointment bias and crossovers.

**Conclusion** This review outlines the development of cohorts to conduct RCTs including the range of use and innovative changes and adaptations. Inconsistencies in the use of terminology and concepts are highlighted. Guidance now needs to be developed to support the design and reporting of RCTs conducted using cohorts.

## STRENGTHS AND LIMITATIONS OF THIS STUDY

⇒ The study's methodological strengths include the use of the CONSORT-ROUTINE extension search strategy to update the scope of work taking place using cohorts.
⇒ Another strength is the overview of methodological writing on trials using cohorts including how informed consent procedures are operationalised.
⇒ While we aimed to assess reporting and terminology, the identification of our sample relied on reporting and terminology.
⇒ The review was limited by the amount of detail available on some important design features, for example, how informed consent processes were operationalised in different clinical populations.
⇒ The studies identified are pre-January 2022. As most are protocols for RCTs yet to run or complete, it is difficult to draw inferences on trends in recruitment and retention. The impact of COVID-19 on these trials is also an unknown factor.

## INTRODUCTION

In medical research, the term cohort designates a group of persons who have one or more characteristics in common.[1] Cohort studies use longitudinal data obtained from a cohort population to investigate the epidemiology of diseases and establish associations between explanatory factors and health outcomes. Randomised controlled trials (RCTs) create treatment groups through random allocation in order to assess the effects of interventions such as screening, surgery or drugs. Traditional RCTs are often expensive and time-consuming, often rendering designs that are of limited utility including small sample sizes, low generalisability and lack of long-term outcomes. There is an increasing need for innovative research designs to overcome these shortcomings.[2–9]

RCTs conducted within cohorts is an innovative approach which uses new or existing cohort infrastructures and cohort study participants to identify and recruit trial participants and collect outcome data, through the regular measurement of cohort outcomes and procedures.[10 11]

Cohort data infrastructures generate and collect data for the purpose of research and RCTs can be conducted using the cohort infrastructure. RCTs can also use administrative data,[12] Electronic Health Records (EHRs),[13] and registry data infrastructures,[14] that is, the classic forms of routine data collection. In this scoping review, we cover RCTs using research-based cohorts, but not RCTs using routine, administrative and registry-based data infrastructures which have been described elsewhere.[12–15]

Compared with traditional RCTs, there are several advantages to using cohorts to conduct RCTs. If an existing cohort study is used, then the time needed for developing trial recruitment and data collection systems can be reduced. RCTs conducted within cohort studies may recruit a more generalisable sample of participants.[16] Recruiting trial participants from a cohort provides information on those who decline to participate in the trial. The cohort can also provide ongoing information as to the natural history of the condition and treatment as usual (TAU) and collect outcomes well beyond the time horizon of many traditional RCTs.

There is variation in how this innovative approach to trial design is named and described. A commonly cited approach is the cohort multiple RCT (cmRCT) design[2]; another frequent and more recent term is the Trials within Cohorts (TwiCs) design.[11] In this design, participants enrol in an observational cohort study with regular outcome measurement. This provides a framework for the implementation of multiple RCTs. For each RCT embedded in the cohort, a random selection of RCT-eligible patients is contacted and offered the intervention. Outcomes of participants randomly allocated to the intervention group are compared with outcomes of RCT-eligible patients not randomly allocated to the intervention, who receive standard/usual care as defined within the cohort. The information provided to potential trial participants and the consents sought are 'patient-centred'. The processes in this design aim to replicate informed consent practices as they would be applied in clinical care, that is, patients are only informed about an intervention if and when they will have access to the intervention.[2]

There is variation in how informed consent is operationalised in trials using cohorts. Some RCTs using cohorts use a standard approach to informed consent. In the standard approach, all potential trial participants are informed of all group allocations and also that group allocation will be determined by chance not choice. This information is provided before randomisation. Other trials conducted using cohorts take different approaches to how and when potential and actual participants are given information and their consent sought. A key feature of the TwiCs design is the staging of informed consent. The origins of this approach aimed to be 'patient-centred'. The 'staged-consent' approach[17] further refines this by seeking explicit consent from everyone recruited at the cohort enrolment stage to not be informed in the future if they are randomised to the usual care control group of an RCT.

The timing of randomisation of eligible cohort participants varies. Many trials randomise at one moment in time, using a single-batch sampling approach in their closed or recruiting cohorts. Some use a sequential process whereby individual patients are randomised as soon as they become eligible, which may or may not coincide with the diagnosis.[18] Given the relative novelty of RCTs using cohorts and the increasing adoption of these designs,[1] there is a need to understand how this approach is being used in health research. This scoping review supplements the development of the Consolidated Standards of Reporting Trials (CONSORT) extension for the reporting of RCTs conducted using cohorts and routinely collected data (CONSORT-ROUTINE),[19] which was developed to guide the reporting of the unique characteristics of trials conducted using both research cohorts and three kinds of routinely collected data (EHR, registry and administrative data).[20]

The primary aim of this review is to examine the extent, range and nature of research activity for RCTs conducted using cohorts. Given the heterogeneous terminology and overlapping design details that became obvious during the development of reporting guidelines,[19] the secondary aim is to clarify working definitions and conceptual boundaries for RCTs using cohorts.

## METHODS

The design and reporting of this review follow the PRISMA Extension for Scoping Reviews guidelines.[21] The protocol is accessible at Open Science Framework: https://osf.io/ke8pw.[22]

A scoping review is defined as 'a form of knowledge synthesis that addresses an exploratory research question aimed at mapping key concepts, types of evidence, and gaps in research related to a defined area or field by systematically searching, selecting, and synthesizing existing knowledge'.[23]

### Search strategy

The search strategy from the original scoping review that was conducted to support the development of the CONSORT Extension for Trials using Cohorts and Routinely Collected Health Data[20 24] was updated to cover January 2007 to December 2021. The original search strategy was designed and conducted by an experienced research librarian familiar with knowledge synthesis related to research methods and reporting. This was in collaboration with the CONSORT project team and peer-reviewed using the Peer Review of the Electronic Search Strategy (PRESS).[25]

Searches were performed to identify publications describing methodology, trial protocols and results from RCTs that were conducted using cohorts. Searches were undertaken in January 2022 in Ovid MEDLINE Epub Ahead of Print, In-Process & Other Non-Indexed

Citations, Ovid MEDLINE Daily and Ovid MEDLINE and EBM Reviews—Cochrane Methodology Registry (Final issue, third Quarter 2012). Searches were conducted covering a 15-year period from January 2007 to December 2021, which allowed the identification of relatively recent RCTs that used cohorts (online supplemental file 1). No language restrictions were applied. Ten subject experts within the field were contacted to identify new protocols for trials that are being planned and other publications that may have been missed.

The references were imported from the database into Refworks, duplicates removed and then transferred into DistillerSR (Evidence Partners, Ottawa, Canada). A coding manual based on eligibility criteria was developed (online supplemental file 2). A pilot test of the coding manual was performed prior to the study's inception.

Titles and abstracts were screened independently by two reviewers. A liberal accelerated method was used to identify articles for inclusion in full-text review, where titles and abstracts were screened by one reviewer (BJN), and excluded articles were screened by a second reviewer (CR).[26] Full texts were screened independently by two reviewers (BJN and PF), and any dissonance was resolved by a third reviewer (CR). At the full-text screening, each reviewer indicated how the cohort was used (used for recruitment, ascertainment of outcomes, both recruitment and outcomes).

### Eligibility criteria
Inclusion criteria: any report of cohorts or results of (a) cohorts that are used (or planned to be used), to conduct RCTs, (b) RCTs within cohorts which use cohorts to recruit participants and/or collect trial outcomes (c) methodological studies discussing the use of cohorts for RCTs (online supplemental file 2).

### Data extraction
Data were extracted into a Google Documents spreadsheet. For articles reporting primary research, the following information was extracted: condition being studied using ISRCTN registry condition categories,[27] age group, setting (primary/secondary care/other), country/continent, intervention(s) planned or received, unit of randomisation (individual/cluster) and comparator.

We also extracted data relating to the use of the cohort (participant identification and collection of outcome data), the terminology used to describe the trial design approach, and the approach used to inform and seek consent. Data extraction was completed by a single reviewer (BJN) and independently validated by a second reviewer (CR).

### Analysis and clarity of reporting
The reporting of this review followed the guidelines of the Preferred Reporting Items for Systematic Reviews and Meta-Analyses extension for scoping reviews (PRISMA-ScR).[21] The results were reported as a mixture of descriptive numerical synthesis and narrative synthesis.

The protocols and results of cohorts and RCTs in cohorts are reported in the tabular analysis and in the narrative synthesis, and the methodological articles are described through narrative synthesis of themes for history and development of the terminology and concepts.

### Patient and public involvement
Although there is no patient and public involvement (PPI) in this scoping review, PPI regarding the design is described in the *Commentary on acceptability of design* section.

## RESULTS
The results of the search are presented in figure 1. Of 2628 potentially relevant citations, 143 were eligible of which 109 articles were suitable for both quantitative analysis and qualitative synthesis, and 34 methodological articles for qualitative synthesis only.

Of the 61 protocols, the majority (62%, 38/61) described planned cohorts and RCTs together. 18% (11/61) were for protocols for cohorts and 20% (12/61) were for RCTs in cohorts.

### Research activity for trials conducted using cohorts
In total, 109 publications were reviewed to assess the extent, range and nature of research activity for trials conducted using cohorts. This included 61 protocols reporting on planned cohorts and/or RCTs, 39 papers reporting RCT results, and 9 papers reporting results of cohort studies established with the facility to host future RCTs. Table 1 reports the characteristics of articles reporting protocols for cohort studies and RCTs in column 1 (green). Articles reporting the results of RCTs embedded in cohorts and the results of cohorts set up hosting RCTs are reported in columns 2 and 3. The references throughout give examples of publications in these three categories. Results are presented as absolute values and percentages.

#### Use of cohort
All included studies used cohorts to identify potential participants and almost all (91%, 99/109) also used the cohort to obtain some or all outcome data. Of the 39 RCT result papers, almost all (92%, 36/39) described trials which had used cohorts for both recruitment and outcome measurement. Only three RCT result papers reported using cohorts just for recruitment.

#### Conditions being studied
This section describes the conditions being studied using ISRCTN registry condition categories.[27]

The most common category was signs and symptoms (population health and risk prevention), covering approximately a third of protocols (21/61), a quarter of articles reporting trial results (10/39) and more than a third of cohort descriptions (4/9). Where there was also a specific disease/condition, the study was also placed under this category. For example, a protocol for a population health COVID-19 cohort[28] was placed within both

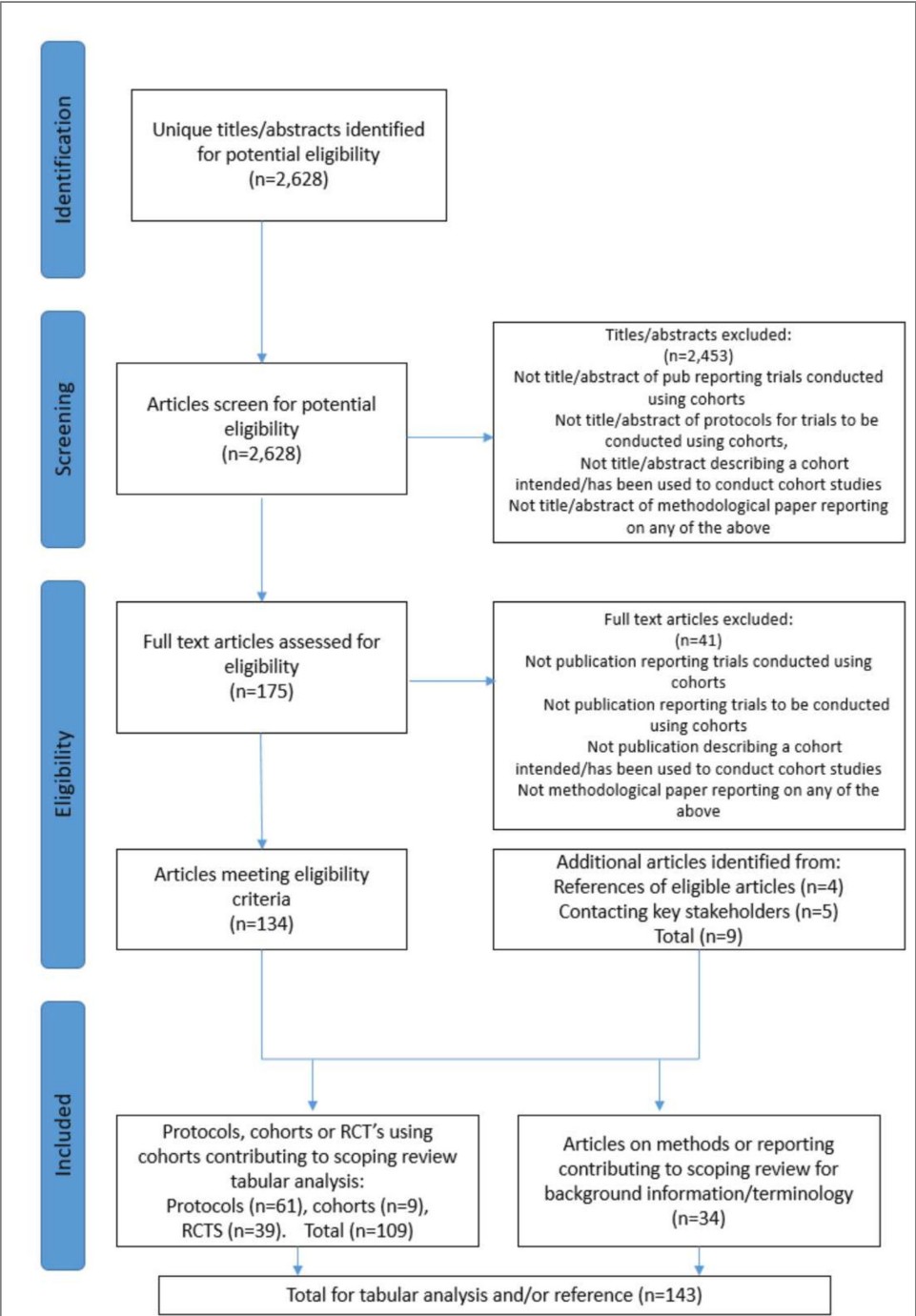

**Figure 1** PRISMA flowchart of the included articles.

the Signs and Symptoms category and the Respiratory category. A cohort that recruited in pregnancy was categorised as Pregnancy and Childbirth as well as Signs and Symptoms/Population Health.[29]

### Age groups
Most studies were of adults – 75% (46/61) of protocols, 87% of RCT results (34/39) and 78% of cohort descriptions (7/9). Studies of children included 13% (5/39) of RCT results, 15% of protocols (9/61) and 11% of cohorts (1/9). Only 10% of protocols (6/61) studied both adults and children. These included planned cohorts for both

parents and children[29 30] and a trial that straddled the child/adult threshold.[31]

### Settings
The most common setting was secondary care with 66% of protocols (40/61), 56% of RCTs (22/39) and 56% of cohorts (5/9). Primary care settings accounted for 20% of protocols (12/61), 31% of RCTs (12/39), and 44% (4/9) of cohorts. The remaining studies described protocols 15% (9/61) and RCTs 13% (5/39) in community[32] and educational[33 34] settings.

**Table 1**  Characteristics of publications reporting protocols for cohorts and results of trials using cohorts

| | No (%) of PROTOCOLS of cohort studies±RCTs (n/61) | | No (%) of RESULTS of RCTs embedded in cohort studies (n/39) | | No (%) of RESULTS of cohort studies set up to host future RCTs (n/9) | |
|---|---|---|---|---|---|---|
| | n= | % | n= | % | n= | % |
| Use of cohort | | | | | | |
| Participant identification | 61 | 100% | 39 | 100% | 9 | 100% |
| Outcome data | 54 | 89% | 36 | 92% | 9 | 100% |
| Condition being studied | | | | | | |
| Signs and symptoms (population health) | 21 | 34% | 10 | 26% | 4 | 44% |
| Musculoskeletal diseases | 12 | 20% | 8 | 21% | 0 | 0 |
| Mental and behavioural disorders | 12 | 20% | 3 | 8% | 0 | 0 |
| Cancer | 8 | 13% | 3 | 8% | 3 | 33% |
| Infections and infestations | 4 | 7% | 3 | 8% | 0 | 0 |
| Respiratory | 4 | 7% | 2 | 5% | 0 | 0 |
| Urological and genital diseases | 3 | 5% | 3 | 8% | 0 | 0 |
| Nutritional metabolic, endocrine | 2 | 3% | 2 | 5% | 0 | 0 |
| Cardiovascular | 1 | 2% | 3 | 8% | 0 | 0 |
| Skin and connective tissue diseases | 1 | 2% | 0 | 0 | 1 | 11% |
| Injury, occupational diseases and poisoning | 1 | 2% | 0 | 0 | 0 | 0 |
| Ear, nose, and throat | 0 | 0 | 0 | 0 | 1 | 11% |
| Pregnancy and childbirth | 1 | 2% | 5 | 13% | 0 | 0 |
| Oral health | 0 | 0 | 2 | 5% | 0 | 0 |
| Other | 0 | 0 | 1 | 3% | 0 | 0 |
| Age group | | | | | | |
| Adults | 46 | 75% | 34 | 87% | 7 | 78% |
| Children | 9 | 15% | 5 | 13% | 1 | 11% |
| Both | 6 | 10% | 0 | 0 | 1 | 11% |
| Setting | | | | | | |
| Secondary care | 40 | 66% | 22 | 56% | 5 | 56% |
| Primary care | 12 | 20% | 12 | 31% | 4 | 44% |
| Other | 9 | 15% | 5 | 13% | 0 | 0 |
| Country/continent | | | | | | |
| Continental Europe | 22 | 36% | 15 | 38% | 4 | 44% |
| United Kingdom and Ireland | 21 | 34% | 13 | 33% | 3 | 33% |
| North America | 7 | 11% | 3 | 8% | 1 | 11% |
| Australia | 5 | 8% | 2 | 5% | 1 | 11% |
| Asia | 3 | 5% | 3 | 8% | 1 | 11% |
| South America | 2 | 3% | 0 | 0% | 0 | 0 |
| Africa | 1 | 2% | 3 | 8% | 0 | 0 |
| Intervention (s) | | | | | | |
| Behaviour change | 17 | 28% | 13 | 33% | N/A | 0 |
| Surgical | 13 | 21% | 5 | 13% | N/A | 0 |
| Drug | 4 | 7% | 12 | 31% | N/A | 0 |
| Psychological approaches | 9 | 15% | 3 | 8% | N/A | 0 |
| Chemo-radiation | 5 | 8% | 2 | 5% | N/A | 0 |
| Complementary therapy | 3 | 5% | 4 | 10% | N/A | 0 |

**Table 1** Continued

| | No (%) of PROTOCOLS of cohort studies±RCTs (n/61) | | No (%) of RESULTS of RCTs embedded in cohort studies (n/39) | | No (%) of RESULTS of cohort studies set up to host future RCTs (n/9) | |
|---|---|---|---|---|---|---|
| | n= | % | n= | % | n= | % |
| Physical/manual therapy | 2 | 3% | 5 | 13% | N/A | 0 |
| Screening | 2 | 3% | 2 | 5% | N/A | 0 |
| Other | 10 | 16% | 5 | 13% | N/A | 0 |
| Comparator | | | | | | |
| Usual care | 50 | 82% | 28 | 72% | 7 | 78% |
| Placebo | 6 | 10% | 6 | 15% | 0 | 0 |
| Unclear/other | 5 | 8% | 5 | 13% | 2 | 22% |

RCT, Randomised controlled trial.

## Country/continent

The majority of publications came from Continental Europe, with 36% (22/61) protocols, 38% (15/39) RCTs and 44% (4/9) stand-alone cohorts. This was closely followed by the UK and Ireland, with contributions from North America, Australia, Asia, South America and Africa.

## Interventions

The most common interventions studied were behaviour change with 28% (17/61) of protocols and 33% (13/39) of RCTs. Some population health-based cohorts planned to identify interventions after the cohort study had started data collection and thus were unable to identify which interventions would be tested (eg,[29 30 35 36]). Some interventions fell into two or more categories, either because they could be classed as both, for example, behavioural change and psychological approaches (eg,[37]) or because the trial combined different categories of interventions, for example, complementary therapy and nutrition (eg,[38]). Some interventions were categorised as 'other', for example, helmet therapy,[39] cough treatment algorithms[40] and nurse-led medicine monitoring.[41]

## Comparators

Most studies included TAU comparators – 82% (50/61) of protocols, 72% (28/39) of RCTs, and 78% (7/9) of cohorts as defined within the cohort. TAU is assumed to refer to unrestricted usual care with no intervention, restriction or standardisation placed on the clinician or recipient, but this was not clarified in some studies. Placebo was used in 10% (6/61) of protocols[42 43] and 15% (6/39) of RCTs.[44–46] The 'other' category included an active comparator.[47]

In several cohorts, it was not stated what the trial comparators would be (eg[48]), but as expected, all RCTs using the staged approach had a usual care comparator. In the results of stand-alone cohorts intended to be used for RCTs, we categorised them as having a usual care comparator if the publication made reference to using a staged or person-centred approach to informed consent.

## Terminology and conceptual boundaries

In addition to publications reporting research activity (table 1), there were 34 methodology papers (table 2). These were analysed to explore the development of the terminology and concepts and contribute to the

**Table 2** Methodological papers

| Focus of paper | Numbers (n/34) | | First author and reference |
|---|---|---|---|
| | N | % | |
| Cohort designs for clinical areas | 6 | 18 | Ahmed,[63] Couwenberg,[18] Gal,[62] Heaven,[74] Young-Afat,[65] Zakrzewska[75] |
| CmRCT/TwiCs design | 17 | 50 | Bibby,[76] Candlish,[77] Clegg,[78] Flory,[79] Kim,[80] Lambin,[81] Pate,[82] Relton,[2 11] Reeves[54] Richards,[83] Van der Velden,[59] Verkooijen,[84] Verweij,[66] Vickers,[85] Viksveen,[16] Young-Afat[17] |
| Learning Health Systems (using TwiCs) | 1 | 3 | Wouters[73] |
| Studies within a Trial (SWATs) using cohorts | 10 | 30 | Arundel,[86] Cockayne,[60] Cotterill,[87] Boyd,[88] Goodwin,[89] Knapp,[90] Ni,[91] Loban,[92] Maruani,[93] Wakabayashi[94] |

CmRCT, cohort multiple RCT; TwiCs, Trials within Cohorts.

**Table 3** Terminology and conceptual boundaries in articles reporting protocols for cohorts, and results of trials using cohorts

| | No (%) of PROTOCOLS of cohort studies±trials (n/61) | | No (%) of RESULTS of RCTs embedded in cohort studies (n/39) | | No (%) of RESULTS of cohort studies set up to host future RCTs (n/9) | |
|---|---|---|---|---|---|---|
| | n= | % | n= | % | n= | % |
| Trial design | | | | | | |
| Individually randomised | 54 | 89% | 34 | 87% | 7 | 78% |
| Cluster randomised | 5 | 8% | 3 | 8% | 0 | 0 |
| Stepped wedge | 2 | 3% | 2 | 5% | 0 | 0 |
| Not known | 0 | 0% | 0 | 0 | 2 | 22% |
| Informed consent approach | | | | | | |
| 'Patient-centred'/'staged-consent' | 31 | 51% | 12 | 31% | 7 | 78% |
| 'All upfront' standard approach | 20 | 33% | 21 | 54% | 0 | 0 |
| Not described | 2 | 3% | 2 | 5% | 0 | 0 |
| Other | 8 | 13% | 4 | 10% | 0 | 0 |
| Design terminology used | | | | | | |
| Cohort multiple RCT (cmRCT) | 14 | 23% | 7 | 18% | 5 | 56% |
| Trials within Cohorts (TwiCs) | 13 | 21% | 5 | 13% | 1 | 11% |
| Cohort-embedded RCT | 9 | 15% | 12 | 31% | 0 | 0 |
| Cohort-nested RCT | 11 | 18% | 4 | 10% | 0 | 0 |
| Other | 14 | 23% | 11 | 28% | 3 | 33% |
| Timing of randomisation to RCT | | | | | | |
| Sequential randomisation | 34 | 56% | 15 | 38% | 0 | 0 |
| Batch randomisation | 20 | 33% | 23 | 59% | 0 | 0 |
| Not stated/known | 7 | 11% | 1 | 3% | 9 | 100% |

cmRCT, cohort multiple RCT; RCT, Randomised controlled trial; TwiCs, Trials within Cohorts.

qualitative synthesis. This table shows the type of trial design, approach to informed consent, terminology used to describe the type of trial design and timing of randomisation to an RCT. The protocols are reported together in column 1 and the results of the RCTs and cohorts for hosting RCTs are given in columns 2 and 3.

### Type of trial design

The majority of protocols and trials reports described individually randomised RCTs. Cluster trial designs were described in 8% (5/61) of protocols and trial reports (3/39) and stepped wedge designs in 3% (2/61) of protocols and 5% (2/39) of trial reports. Table 3 reports the characteristics of articles reporting protocols for cohort studies and RCTs in column 1 (green). Articles reporting the results of RCTs embedded in cohorts and the results of cohorts set up hosting RCTs are reported in columns 2 and 3. The references throughout give examples of publications in these three categories. Results are presented as absolute values and percentages.

### Approach to informed consent

Some studies stated whether they were using either a 'patient-centred' or 'staged' approach to informed consent, either by using these terms or by citing Young-Afat[17] 'Staged consent' publication or Relton[2] when describing their approach.

Approximately half (51%) of protocols (31/61) and 31% (12/39) of trials reported using a staged approach. Studies not using this approach included cluster trials.[34 49–52]

### Terminology

The term cmRCT was used in 23% (14/61) of protocols and 18% (7/39) of RCTs. TwiCs was used in 21% (13/61) of protocols and 13% (5/39) of trials. Cohort-embedded RCT was used in 15% (9/61) of protocols and 31% (12/39) of trials. Other terms used include cohort-nested RCT, cohort RCT and cohort study.

In relation to informed consent, there were 68 publications that used or described 'patient-centred' or staged approaches. Publications were identified if they either (a) used the term 'patient-centred' or 'staged' or cited Young-Afat[17] or Relton[2] when describing their approach. One study cites using a staged approach but does not use any terminology to describe the design of its future RCTs.[53] The terminology used in the 67 remaining publications is reported in table 4.

**Table 4** Terminology used to describe trial designs for randomised controlled trials (RCTs) with a patient-centred/staged approach to IC

| Trial design/year | 2010–2011 | 2012–2013 | 2014–2015 | 2016–2017 | 2018–2019 | 2020–2021 |
|---|---|---|---|---|---|---|
| cmRCT | 2 | 84 | 78 81 83 | 16 17 74 75 77 79 82 95 | 54 | |
| | 35 96 | 67 97 | 31 98–100 | 69 101–103 | 104 105 | |
| | | 61 | 106 | 58 107 | 108–110 | |
| TwiCs | | | | 11 | 54 76 80 | 18 65 66 73 111 112 |
| | | | | 29 | 33 113–115 | 30 68 70 116–120 |
| | | | | | 37 38 121 | 71 122 |
| Cohort-embedded/nested RCT | | | 63 'em' | | | |
| | | 123 'n' | 124 'n' | 40 'n' | 64 'em' | |

Methodological literature (n=24); Protocol (n=31); RCTs using cohorts (n=12).
cmRCT, cohort multiple RCT; TwiCs, Trials within Cohorts.

Table 4 reports the terminology of studies using staged-informed consent in methodological publications and publications reporting research activity. This table shows the terminology for methodological literature, protocols and RCTs using cohorts. The reference to each publication is given and the colour index is given below the table. The terms cmRCT, cohort-embedded and cohort-nested are given from 2010 when the patient-centred approach to IC was first described, and the term TwiCs from 2016 when it was first coined in the literature. Cohort-embedded and cohort-nested are given together with the codes 'em' for embedded and 'n' for nested.

From 2018 onwards, there is a noticeable shift in the terminology used in protocols, with the majority of protocols describing TwiCs, a name that was coined at the 2016 Ethics of TwiCs Symposium.[11] This shift in terminology was also seen in the methodological literature, where 100% (9/9) papers identified from 2018 to 2021 refer to the design as TwiCs, with one paper using both terms TwiCs and cmRCT.[54]

### Timing of randomisation

56% (34/61) of the protocols and 38% (15/39) of the RCTs stated that they would or had used sequential randomisation, compared with batch randomisation which was used by 33% (20/61) of protocols and 59% (23/39) of RCTs. The timing of randomisation was not clear in 11% of protocols and 3% of RCTs.

### DISCUSSION

Three main areas emerged from the narrative synthesis of themes arising in the studies and methodological literature: commentary on efficiency of recruitment, commentary on acceptability and new conceptual developments. These are discussed below.

### Commentary on efficiency of recruitment

Unusually, no trials reported being unable to recruit to time and target.[55]

Most articles commented on the efficiency of the design, enhanced recruitment and the potential to embed in multiple RCTs. Young-Afat et al[56] reported that 88% of patients invited to join the UMBRELLA cohort consented to participate, most of whom (88%) gave broad consent for randomisation to future interventions. Coebergh van den Braak et al[57] proposed that the efficiency of this approach arises from physicians saving time by only explaining an intervention that they can *actually* offer to a patient. Viksveen et al[58] reported that the trial 'over-recruited', and how the design enabled the acceptability of an intervention to be measured by intervention take-up in the intervention group. Viksveen et al[16] also argued that this approach has the potential to recruit more representative populations than stand-alone recruitment to an RCT and that the 'main benefits of using the cmRCT design to test an intervention for self-reported depression were full, fast and efficient recruitment; lower attrition rates than other depression trials; and a trial population broadly similar to the general population of patients self-reporting chronic moderate to severe depression'.[16]

Most trials (33/39) and all protocols stated that they had or would use an Intention to Treat (ITT) analysis as part of their statistical approach. Some also recommended using a complier average causal effect (CACE) analysis instead of per protocol analysis due to risk of bias.[16 58] In CACE analyses, outcomes in patients who take up the offer of treatment are compared with outcomes in the no offer group of patients who would have taken up the offer had they received it. Some authors used instrumental variables (IV) analysis, as this type of CACE analysis takes into account patients' baseline characteristics,[16 37 59–61] and commented that this analysis was appropriate where

there is lower take-up in the intervention arm, especially where the intervention is the offer of treatment.

One publication (Gal *et al*[62]) compared recruitment to a stand-alone RCT to recruitment from within a stage-consenting cohort. Recruitment from the stage-consenting cohort was significantly cheaper and faster and prevented contamination between groups 'but non-compliance due to refusal of the intervention was higher compared with conventional pragmatic exercise oncology RCTs, which may dilute the estimated intervention effect'.

### Commentary on acceptability of design

Several studies described PPI) work around the trial design in the field of prostate cancer research[63] and kidney disease.[64] An online survey conducted in collaboration with the charity Kidney Cancer UK to ascertain acceptability of the study's design and the willingness of patients to be recruited into the cohort found 'strong support and need from the kidney cancer patient community for the broad concepts of the study'.

The first published evaluations of broad informed consent for randomisation within a TwiCs study were conducted in the Netherlands.[65 66] Cancer patients participating in ongoing TwiCs[65] and a separate study were surveyed between two to six years after cohort entry.[66] The approach was found to be generally well accepted by patients with only 2% of patients in the trial usual care control group stating that they would feel negative about not being selected for interventions while their data are being used as a comparator. Both evaluations support the use of the TwiCs design with the staged-informed consent procedure.

### New conceptual developments

Most cohorts using the TwiCs design approach were of clinical populations. Some publications described population cohorts, for example, England's 'Born in Bradford Better Start' (BIBBS) cohort[29] and the Australian 'Generation Victoria' population cohort.[30] Some publications described 'multi-national' cohorts – the Scleroderma Patient-centred Intervention Network (SPIN) cohort which involves 42 centres globally,[67] the Emotional Competence in Young Adults (Ecoweb) PROMOTE platform to promote mental health in young adults,[68] which uses a web-based multinational recruitment strategy, and the European Prevention of Alzheimer's Dementia Longitudinal Cohort Study (EPAD LCS).[53] The EPAD Project is a large collaboration supporting a platform for the testing of multiple interventions. Solomon *et al* describe how "As the EPAD project is multistaged, staged consent will be used as a decision-making model. Staged consent feeds relevant information—bit by bit, extended over time—to participants and study partners, and asks informed consent at every step when they need to make important decisions".[53]

This review reports the characteristics of trials conducted using cohorts and observed how terminology and conceptual boundaries evolved over time in the methodological literature and the literature reporting research activity.

A diverse range of cohorts and trials using cohorts in a wide range of populations were identified. The TwiCs design approach is being used for a wide range of clinical conditions and interventions. A major clinical group is oncology[69–72] in the Netherlands. Use of the TwiCs design is spreading with innovative work taking place in a range of locations and new collaborations are leading to the development of multi-national recruitment.

The term 'staged' approach to informed consent coined in 2016[17] has replaced the earlier term 'patient-centred' in the majority of studies.

All the RCTs using the TwiCs design had usual care for at least one of the comparators – a prerequisite to using a staged approach, that is, not to contact cohort participants further in the usual care group if they are not selected for trial interventions.

Either batch or sequential randomisation can be used as part of a TwiCs design depending on clinical condition and demands of the study. However, the majority of protocols use sequential randomisation, where individuals are randomised as they become eligible for the experimental intervention.[18]

The TwiCs approach has been cited as an example of an 'Optimisation learning healthcare system'[73] for encompassing strategies that contribute to a healthcare system in which 'the process of generating and applying the best evidence will be natural and seamless components of the process of care itself.' Use of a staged-informed consent process, routine data and individual randomisation at the point of diagnosis could all be considered examples of these strategies.

To date the acceptability work around the TwiCs design has been mainly conducted in the field of oncology. Further work to assess the impact for different clinical groups is recommended. There is also a lack of direct comparison with standard RCTs.

This review confirms findings in the earlier scoping review conducted by the CONSORT-ROUTINE team[19] that there is heterogeneity in the terminology and conceptual boundaries for the use of cohorts for RCTs. The term TwiCs is now used for the majority of studies but there is still some variation in terminology and use of the design. There are no current guidelines to support consistency of use.

### Methodological limitations

While we aimed to assess reporting and terminology, the identification of our sample relied on reporting and terminology. The review was limited by the amount of detail available on some important design features for example, how informed consent processes were operationalised in different clinical populations.

As all the studies identified are pre-January 2022 and most are protocols rather than completed studies, it is

difficult to draw inferences on trends in recruitment and retention. Moreover, the impact of the COVID-19 pandemic on these trials is unknown.

## CONCLUSION

This review identified increasing sophistication and maturity in RCTs conducted within cohorts in a wide range of conditions and research studies including 'mega-cohorts'.[30] Our findings support the increased use of the TwiCs design to address some of the barriers to recruitment and retention for standard RCTs with usual care comparators. Further research into the acceptability, use and efficiency of staged approaches to informed consent, compared with standard approaches, is required. We recommend the use of the term TwiCs to unify approaches and call for an international symposium to define terminology and develop guidelines for best practice for studies using the TwiCs approach.

**Author affiliations**
[1]Centre for Evaluation and Methods, Wolfson Institute of Population Health, Queen Mary University of London, London, UK
[2]Research Center for Clinical Neuroimmunology and Neuroscience Basel (RC2NB), University Hospital Basel, Basel, Switzerland
[3]Department of Clinical Research, University of Basel, Basel, Switzerland
[4]Meta-Research Innovation Center at Stanford (METRICS), Stanford University, Stanford, California, USA
[5]Meta-Research Innovation Center Berlin (METRICS-B), Berlin Institute of Health, Berlin, Germany
[6]Department of Family Medicine, Western University, London, Ontario, Canada
[7]Institute for Clinical Evaluative Sciences, Toronto, Ontario, Canada
[8]National Perinatal Epidemiology Unit, Clinical Trials Unit, Nuffield Department of Population Health, University of Oxford, Oxford, UK
[9]Center for Research on Population and Health, Faculty of Health Sciences, American University of Beirut, Ras Beirut, Lebanon
[10]Bone and Joint Health, Blizard Institute, Queen Mary University of London, London, UK
[11]Barts Health NHS Trust, Royal London Hospital, London, UK
[12]Centre for Primary Care, Wolfson Institute of Population Health, Queen Mary University of London, London, UK
[13]University Medical Centre Utrecht, Utrecht, The Netherlands
[14]University of Utrecht, Utrecht, The Netherlands
[15]University of Toronto Institute of Health Policy Management and Evaluation, Toronto, Ontario, Canada
[16]Independent Researcher, Basel, Switzerland

**Acknowledgements** We thank Margaret Sampson, MLIS, PhD, AHIP (Children's Hospital of Eastern Ontario, Ottawa, Canada) for developing the search strategies on behalf of the CONSORT team. We thank Dr Philippa Fibert, St Mary's University, Twickenham, London for her help in screening publications for inclusion.

**Contributors** The conception and design of this work, acquisition of data, initial analysis and preparation of each draft have been the responsibility of BJN as part of her PhD studies, supported by CR, XLG and RS. CR, LH, MZ, SE, SJMC, XLG, RS, HMV, JLM and KAMC have contributed substantially to the analysis and interpretation of the work, reviewing each draft critically for important intellectual content. BJN, CR, LH, MZ, SE, SJMC, XLG, RS, HMV, JLM and KAMC are jointly responsible for final approval of the version to be published and agreement to be accountable for all aspects of the work in ensuring that questions related to the accuracy or integrity of any part of the work are appropriately investigated and resolved. BJN is acting as guarantor of the work, accepting full responsibility for the work, had access to the data, and controlled the decision to publish.

**Funding** We have not received a specific grant from any funding agency in the public, commercial or not-for-profit sectors to conduct this research. Publication

costs for this article have been met by Queen Mary University of London and the University of Western Ontario under the BMJ Open Access publication agreement.

**Competing interests** Publication costs for this article have been met by Queen Mary University of London and the University of Western Ontario under the BMJ Open Access publication agreement.

**Patient and public involvement** Patients and/or the public were not involved in this scoping review. PPI regarding the design is described in the *Commentary on acceptability of design* section.

**Patient consent for publication** Not applicable.

**Ethics approval** Not applicable.

**Provenance and peer review** Not commissioned; externally peer reviewed.

**Data availability statement** The electronic search strategies and inclusion and exclusion criteria are uploaded as supplemetary information. References to all studies using a staged approach to informed consent are given in Table 4. A full data set is available on reasonable request.

**ORCID iDs**
Beverley Jane Nickolls http://orcid.org/0000-0002-9844-2493
Lars Hemkens http://orcid.org/0000-0002-3444-1432
Stephen J McCall http://orcid.org/0000-0003-0078-7010
Xavier Luke Griffin http://orcid.org/0000-0003-2976-7523
Jonathon L Maguire http://orcid.org/0000-0002-4083-8612

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
