## [Reviewer comments · BMJ Open]

ARTICLE DETAILS

TITLE (PROVISIONAL)	Randomised trials conducted using cohorts: A scoping review
AUTHORS	Nickolls, Beverley; Relton, Clare; Hemkens, Lars; Zwarenstein, Merrick; Eldridge, Sandra; McCall, Stephen; Griffin, Xavier; Sohanpal, Ratna; Verkooijen, Helena; Maguire, Jonathon; McCord, Kimberly

VERSION 1 – REVIEW

REVIEWER	Hefner, Kathryn The Emmes Company LLC
REVIEW RETURNED	19-Jul-2023

GENERAL COMMENTS	This scoping review covers a timely and interesting topic, and I learned more about current terminology and trends regarding cohorts within RCTs. One limitation that felt glaringly missing from the discussion section, was that the majority (~60%) of the trials were protocols that had not yet been implemented and/or studies completed, so any summary of trends regarding recruitment, retention, etc. are not relevant to the majority of studies in the review. Moreover, given the timing of these planned protocols and study implementation, it is likely that COVID-19 impacted many of these trials, causing delays and/or recruitment challenges, which is in turn likely to contribute to changes to planned study design, including the use or execution of cohorts. So any conclusions about these planned studies should be advanced as tenuous given that my understanding is that they reflect a snapshot in time of the plans for these studies.
--

REVIEWER	Williams, Norman University College London, UCL Division of Surgery & Interventional Science
REVIEW RETURNED	29-Oct-2023

GENERAL COMMENTS	This is a very good piece of work that will be of interest to many readers of this journal.
---

VERSION 1 – AUTHOR RESPONSE

Reviewer 1:

'This scoping review covers a timely and interesting topic, and I learned more about current terminology and trends regarding cohorts within RCTs. One limitation that felt glaringly missing from the discussion section, was that the majority (~60%) of the trials were protocols that had not yet been

implemented and/or studies completed, so any summary of trends regarding recruitment, retention, etc. are not relevant to the majority of studies in the review. Moreover, given the timing of these planned protocols and study implementation, it is likely that COVID-19 impacted many of these trials, causing delays and/or recruitment challenges, which is in turn likely to contribute to changes to planned study design, including the use or execution of cohorts. So any conclusions about these planned studies should be advanced as tenuous given that my understanding is that they reflect a snapshot in time of the plans for these studies'.

Action taken:

We have added a section describing these limitations to the end of the Discussion section. Please see above.

We have also added these limitations to the strengths and limitations section which now reads as follows:

- The study's methodological strengths include the use of the CONSORT- ROUTINE extension search strategy to update the scope of work taking place using cohorts.
- Another strength is the overview of methodological writing on trials using cohorts including how informed consent procedures are operationalized.
- While we aimed to assess reporting and terminology, the identification of our sample relied on reporting and terminology.
- The review was limited by the amount of detail available on some important design features e.g. how informed consent processes were operationalised in different clinical populations.
- The studies identified are pre-January 2022. As most are protocols for RCTs yet to run or complete, it is difficult to draw inferences on trends in recruitment and retention. The impact of COVID-19 on these trials is also an unknown factor.

Reviewer 2:

'This is a very good piece of work that will be of interest to many readers of this journal'.

Action taken:

Thank you. No further action required.